# MTTfireCAL Package for R—An Innovative, Comprehensive, and Fast Procedure to Calibrate the MTT Fire Spread Modelling System

Bruno A. Aparício *, Akli Benali, José M. C. Pereira and Ana C. L. Sá †

Forest Research Centre and Associate Laboratory TERRA, School of Agriculture, University of Lisbon, Tapada da Ajuda, 1349-017 Lisbon, Portugal
* Correspondence: bruno.a.aparicio@gmail.com
† Current address: ForestWISE-Laboratório Colaborativo para a Gestão Integrada da Floresta e do Fogo, Quinta de Prados, 5001-801 Vila Real, Portugal.

**Abstract:** Fire spread behavior models are used to estimate fire behavior metrics, fire hazard, exposure, and risk across the landscape. One of the most widely used fire spread models is the minimum travel time (MTT), which requires a very time-consuming, interactive, trial-and-error calibration process to reproduce observed fire regimens. This study presents the MTTfireCAL package for R, a tool that enables fast calibration of the MTT fire spread models by testing and combining multiple settings and then ranking them based on the model's capacity to reproduce historical fire patterns, such as fire size distribution and fire frequency. Here, we explain the main methodological steps and validate the package by comparing it against the typical calibration procedures in two study areas. In addition, we estimate the minimum number of fire runs required to ensure a reliable calibration. Overall, the use of MTTfireCAL R package and the optimization of the number of ignitions used allowed for a faster calibration of the MTT modeling system than the typical trial-and-error calibration. The MTT modeling system calibrated using MTTfireCAL was also able to better reproduce the historical fire patterns. This tool has the potential to support the academic and operational community working with MTT.

**Keywords:** minimum travel time; fire spread model; calibration; R package; FlamMap





## 1. Introduction

Fire behavior is defined as the "manner in which fuel ignites, flame develops, and fire spreads and exhibits other related phenomena as determined by the interaction of fuels, weather, and topography" [1]. Studying and understanding fire behavior is considered to be a key aspect to achieve fire management goals [2], and is often assessed using fire behavior models. These models estimate metrics, such as rate of spread and fireline intensity, among other variables. One of the most widely used fire spread modeling systems is the minimum travel time (MTT) [3]. The MTT modeling system is included in the FlamMap fire mapping and analysis system [4]. The MTT algorithm calculates two-dimensional fire growth by searching for the pathways with minimum spread time from the cell corners [3]. Unlike FlamMap Basic, which estimates fire behavior independently in all landscape pixels, the MTT algorithm estimates fire behavior resulting from an ignition point and is dynamically influenced by weather and fuels for each simulation time step. The algorithm estimates rate-of-spread using Rothermel's equation [5] and fire intensity using Byram's equation [6], which is then converted to flame length.

The MTT modeling system, and its command-line version FConstMTT, has been widely used to model fire spread and estimate fire behavior in several fire-prone countries, including the USA [7], Portugal [8], Spain [9], Italy [10], Greece [11], and Iran [12]. MTT

has been used in the past with multiple research objectives, from characterizing fire behavior in the landscape (e.g., [8]), quantify the effect of different fuel reduction strategies (e.g., [13,14]), assess economic losses (e.g., [15]), prioritize areas to treat [16], and to support the development of multi-objective fire management strategies (e.g., [17,18]).

MTT fire spread models require a landscape file containing grid data of topographic and fuel characterization of the study area, ignition points that set the start of the fire spread, and weather conditions for the fire spread. Afterward, the MTT algorithm needs to be calibrated to ensure that the estimated fire patterns are reliable [19]. Failing to do so may lead to errors in reproducing key fire descriptors, such as burn probability [20], ultimately undermining the use of fire simulation for research and management purposes.

The calibration of MTT is often done by comparing the historical fire size distribution with the simulated fire size distribution (e.g., [21]) and by correlating the historical fire frequency with the estimated burn probability (e.g., [22]). The calibration process may be divided into four main steps: (i) characterization of environmental conditions associated with wildfires; (ii) adjustment of maximum simulation time (or duration), i.e., the duration that a fire spreads in the landscape; (iii) fire simulation; and (iv) evaluation of the results. The initial step of characterizing the environmental conditions for the study area includes compiling topographic data, surface and canopy data, and the prevailing weather conditions during active fire spread. After characterizing the environmental conditions, the user needs to ensure that predictions reproduce historical fire patterns. This is done by adjusting the maximum simulation time parameter, often tuned using a time-consuming trial-and-error process. This task becomes even more challenging when considering that datasets of time-stamped fire perimeters do not necessarily correspond to the observed duration of active fire spread [23]. Additionally, in the process of replicating the historical fire pattern, multiple values of maximum simulation time (hereafter, duration classes) may be needed, which exponentially increases the complexity of the trial-and-error calibration, and consequently, the time consumed in this step.

Another important time-consuming step during the calibration is the fire simulation itself. To produce reliable estimates of fire spread descriptors, the landscape is usually saturated with thousands of ignitions (e.g., [24]). Generating such a large number of ignitions requires significant computational time and resources. However, this large number of ignitions may not be necessary during the calibration process as none of the fire behavior metrics are used in the calibration process, other than fire size distribution and spatial patterns. Hence, for calibration purposes it is possible that simulating fewer fire ignitions will result in a similar parameterization when compared with saturating the landscape, possibly allowing the user to save time.

The MTT calibration is time-consuming and challenging, particularly for new users. Under the current context of climate change and the expansion of severe fire seasons to new latitudes [25], it is expected that users without prior experience in fire spread modeling will resort to fire spread models, such as MTT (and FConstMTT). Hence, new tools that assist and guide users in the calibration process are of particular interest. Here, we present a new framework that tackles the three major time-consuming steps when calibrating the MTT algorithm: (1) characterization of the environmental conditions driving fire spread; (2) definition of the fire spread duration parameter(s) (trial-and-error process); and (3) time required for fire simulation. We developed the MTTfireCAL package for R [26], an open semi-automatic tool that can significantly decrease the time required to calibrate the MTT algorithm. Specifically, our study aims to (1) present and describe MTTfireCAL, demonstrating how it can be used by applying it to one study area; (2) validate the MTTfireCAL by applying it to two other case studies and by comparing the calibrated duration parameters using the MTTfireCAL package against the traditional trial-and-error procedure; (3) analyze the number of ignitions needed to reproduce the historical fire patterns, and compare it with the classical "landscape saturation" approach; and (4) quantify the time reduction in the calibration process obtained by using the MTTfireCal. The package can be downloaded from GitHub at https://github.com/bmaparicio/MTTfireCAL (accessed on 21 May 2023).

## 2. Methods

### 2.1. Study Area

The MTTfireCAL was applied to three different study areas in Portugal where fire spread models were previously calibrated: Barlavento Algarvio, Médio Tejo, and Área Metropolitana do (AM) Porto (Figure 1). AM Porto has 92,590 ha and the fire regime is characterized by a combination of infrequent but large and intense wildfires, mostly in shrublands and pine and eucalyptus forests; small and low-intensity fires in the wildland-urban interface; and some winter shrubland fires related with pastoralism [27]. Médio Tejo has an area of 350,450 ha and the fire regime is a mixture between infrequent but large and intense wildfires and small wildfires in agriculture and agroforestry areas. The Barlavento Algarvio has 242,214 ha and is characterized by infrequent but large and intense wildfires that burn mainly pine and eucalypt forests [27].

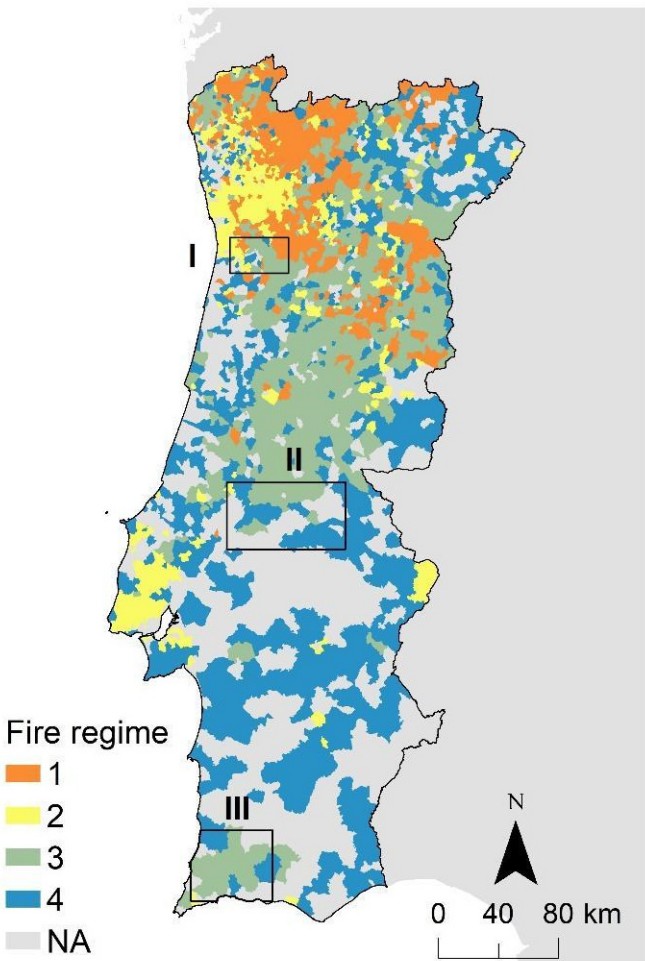

**Figure 1.** Location of the three study areas used to calculate the minimum number of fire runs required for calibration and for the validation of the "MTTfireCAL" package. AM Porto is identified as zone I, Médio Tejo is identified as II, and Barlavento Algarvio is identified as III. Fire regime 1: winter shrubland fires related to pastoralism; Fire regime 2: high incidence of fire events with small areas burned, mostly located in regions with high population density; Fire regime 3: infrequent but large and intense wildfires, mostly in shrublands and pine and eucalyptus forests; Fire regime 4: small wildfires in agriculture and agroforestry areas. Adapted from [27].

The Barlavento Algarvio region is used throughout the manuscript to illustrate the application of MTTfireCAL. The remaining two study areas were used to compare the outcomes of a traditional trial-and-error calibration process against the semi-automatic calibration using the MTTfireCAL (Sections 2.8 and 3).

### 2.2. Flowchart of MTTfireCAL

The workflow of MTTfireCAL is shown in Figure 2. It has eight functions developed to calibrate the MTT fire spread modeling system: *get_fire_weather*, *fire_weather_nc*, *build_report*, *gen_ign*, *run_fconstmtt*, *run_fconstmtt_simple*, *evaluate_fire_size*, and *evaluate_BP_nxburned*. Each function (or set of functions) is responsible for a key step in the calibration process, as described in detail in the next sections. There are five mandatory input files to use the MTTfireCAL: a shapefile of the study area, a shapefile with dated historical fire perimeters, a grid of ignition probability, one (or more) grid(s) of fuel models, and one (or more) landscape file(s). The remaining data required to run the MTT fire spread model (e.g., weather conditions, fms files, ignition points, etc.) are generated within the package. However, the users may also use their own data. The inputs are explained in detail in Section 2.3.

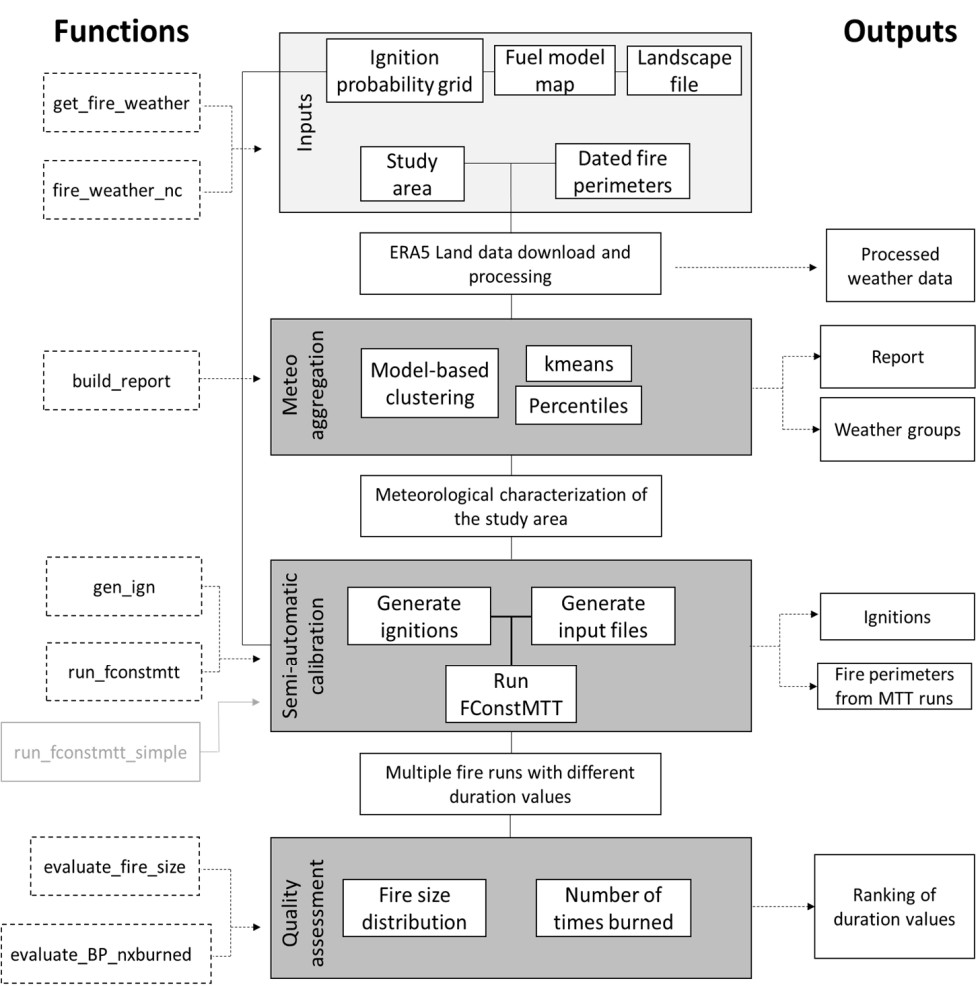

**Figure 2.** Workflow of 'MTTfireCAL' package for R. Each dashed box corresponds to a function that is used in different steps of the calibration. The initial inputs required are listed in the central box in light grey. The objective of each function is identified in the central panel in grey boxes. The most important outputs generated from each function are identified in the rightmost panel. The function *run_fconstmtt_simple* (in grey) can be used independently from the other functions.

The study area and dated historical fire perimeters are used in the function *get_fire_weather*, which downloads ERA5-Land [28] or ERA5 [29] meteorological data from the Copernicus Climate Data Store (CDS). The meteorological data is processed and stored in a text file that will later be used in other functions to characterize the historical fire weather conditions. The function *fire_weather_nc* is similar to the previous function but instead of downloading the meteorological data, it uses NetCDF files that were previously downloaded, processing the meteorological in the same manner as *get_fire_weather*. This

function is particularly useful in cases where an unstable internet connection or saturation of the CDS may delay data download or cause unexpected errors.

The function *build_report* uses the processed weather data, the shapefile of the study area, and the dated historical fire perimeters to classify the weather data into weather groups. This function outputs the meteorological classification of the fire weather conditions associated with the historical wildfires considered, and a report with the characterization of the historical fire size distribution. An example of the automatic report is shown in Supplementary Materials Section S7.9.

After characterizing the meteorology in the study area, the function *gen_ign* can be used to generate ignitions. This function requires two input files that are generated outside the MTTfireCAL package: a grid of ignition probability and a fuel model map. Once the ignitions are created, the function *run_fconstmtt* is used to run FConstMTT with multiple combinations of the duration parameter(s), weather scenarios, and fuel models. This function also requires a landscape file generated outside the package. As an alternative to the function *run_fconstmtt* which depends on the previous functions listed, the function *run_fconstmtt_simple* uses one set of weather conditions set by the user to test multiple combinations of the duration values.

Finally, the last key step of MTTfireCAL is the quality assessment of the fire spread simulations. The function *evaluate_fire_size* compares the historical and simulated fire size distribution, while the function *evaluate_BP_nxburned* compares the simulated burn probability with the historical fire frequency. The combinations of fire spread durations are ranked by their goodness-of-fit in reproducing historical patterns (fire size distribution and fire frequency) using multiple performance metrics, as explained in Section 2.8.

*2.3. Data Required*

2.3.1. Dated Historical Fires

MTTfireCAL uses dated historical fire perimeters to characterize the weather conditions during a fire event in the study area. This shapefile contains information on the start and end dates for each fire event, which can be obtained from national/regional databases and/or using satellite data [23]. In the absence of national or regional data, the user may use global time-stamped fire perimeters (e.g., [30,31] to create the input shapefile. For the task of characterizing the historical fire size distribution in the study area, dated or undated fire perimeters may be used.

2.3.2. Study Area Boundaries

The shapefile of the study area is used to select the historical fire perimeters that will be used throughout the calibration process. The shapefile must contain only one feature (i.e., one polygon). The user should also consider that the shapefile of the study area must be obtained by generating a buffer surrounding the area of interest to take into account the transmission of wildfires from surrounding areas to the study area (e.g., [32]).

2.3.3. Fire Weather

The days of fire spread identified in the fire perimeter shapefile are used as input for the *wf_request* function (ecmwfr R package—[33]). This function allows us to automatically download hourly weather variables from the ERA5-Land [28] or ERA5 [29] dataset from CDS. The MTTfireCAL downloads hourly 2 m temperature, 2 m dewpoint temperature, and 10 m of u- and v-components of wind. The ncdf4 package [34] is then used to process the downloaded data. The 2 m temperature and the 2 m dewpoint temperature are combined using the August–Roche–Magnus formula [35] to estimate the relative humidity. The wind speed and wind direction are calculated from the 10 m u-component and v-component. All weather variables are then converted to the International System of Units (km/h for the wind speed, degrees for wind direction, degrees Celsius for temperature, and percentage for the relative humidity) and saved as a text file.

The ERA5-Land reanalysis dataset has been shown to be a valid data source of meteorological variables [36,37]. Notwithstanding, local data can be used whenever available (e.g., from a local weather station(s)). If that is the case, then the data must have the same format as the fire weather produced automatically (see example in Table S1) and should be specified as an input in a later stage (see Section 2.4).

### 2.3.4. Ignition Probability

The ignition location is an essential input to estimate fire spread and behavior descriptors, particularly burn probability [20], as it sets the starting point of the fire spread. The location of ignitions used to simulate fire spread is derived from an ignition probability surface that reproduces the broad historical spatial ignition patterns in the study area. Usually, the ignition probability surface is created from the historical ignition points by creating a smooth grid using a fixed search distance (e.g., kernel density; [22]).

### 2.3.5. Map of Fuel Models

A map of surface fuel models is essential to run any fire spread simulation. Fuel models quantitatively describe major groups of vegetation that are responsible for surface fire propagation (e.g., litter, herbs, shrubs, slash; [5]). If custom fuels are used (e.g., [38–40]), a fuel model file containing their parameterization is required (.fmd file). The map of fuel models is part of the generated landscape file (created outside the MTTfireCAL) and is also necessary to generate the fire ignitions in the landscape by ensuring that ignition locations are restricted to burnable areas.

### 2.3.6. Landscape File

The landscape file represents a multi-layer raster format composed of elevation, slope, aspect, fuel models, and canopy cover. It can also include crown-related variables: stand height, canopy base height, and canopy bulk density. The landscape file can be generated in the software FlamMap [4].

### 2.4. Fire Weather Data and Classification (Functions get_fire_weather, fire_weather_nc, and build_report)

The function *get_fire_weather* automatically downloads the required weather variables from ERA5-Land dataset. The downloaded weather data is grouped using the *build_report* function. This results in the creation of weather scenarios that are used in the fire spread simulations. Weather data can be grouped using percentiles or using cluster classification. A cluster classification algorithm groups the hourly weather data associated with multiple historical fires using similarity measures. Fire weather data are classified into clusters whose centroids are daily averaged values of temperature, relative humidity, and wind speed. Then, the frequency of each wind direction is calculated for each meteorological cluster. Alternatively, the percentiles classification uses hourly weather data to calculate the percentiles of the variables temperature, relative humidity, and wind speed. For temperature and wind speed the 95th, 50th, and 25th percentiles are computed, and for relative humidity the 5th, 50th, and 75th percentiles of the relative humidity are calculated.

The user may also define the active period of fire spread to subset the interval of hours that will be used in the creation of weather groups. For instance, exploratory analysis using satellite data shows that the energy released by wildfires in Portugal is highest between 14 h and 22 h (Figure S1).

When clustering classification is chosen as the method to create weather groups, MTTfireCAL uses two algorithms: K-means classification [41] and model-based clustering classification [42]. K-means classification is an iterative algorithm that partitions the dataset into K pre-defined distinct non-overlapping clusters. It assigns observations to a cluster such that it minimizes the sum of the squared distance between the data points and the cluster's centroid (arithmetic mean). A lower variation represents a more homogeneous cluster [43,44]. After the creation of the K clusters, the elbow method and Silhouette scores

are exported so that the user can select the optimal number of clusters. Nonetheless, the interpretation and the choice of cluster solution are often subjective [42]. The K-means classification is calculated in MTTfireCAL using the *factoextra* R package [45].

Model-based cluster analysis (MBCA) was designed for modeling an unknown distribution as a combination of simpler distributions [46]. In this classification method, the optimal number of clusters is automatically calculated by fitting a finite mixture model to the fire weather database using the Bayesian information criterion selection [8]. It also produces the clusters' geometric features [47]. The model-based clustering is calculated in MTTfireCAL using the *mclust* R package [48].

After running the weather data classification, the *build_report* function produces two matrices of frequencies that summarize the historical fire weather conditions. The first identifies the centroid values of temperature, relative humidity, and wind speed of each cluster, and the relative frequency of each cluster (Table 1).

**Table 1.** Example of the centroid's values of each cluster for the meteorological conditions driving fires produced by the function *build_report*. The column cluster represents the cluster id, T represents temperature (°C), RH represents relative humidity (%), WS represents wind speed (km/h), and RF represents the relative frequency of each cluster.

| Cluster | T | RH | WS | RF |
|---------|-----|-----|-----|------|
| 1 | 34 | 24 | 20 | 0.59 |
| 2 | 27 | 40 | 27 | 0.41 |

The second matrix characterizes the frequency distribution of the wind direction for each fire weather cluster (Table 2).

**Table 2.** Example of a matrix of frequency distribution of the wind direction inside each fire weather cluster produced by the function *build_report*.

| Cluster ID | Wind Direction | Relative Frequency (%) |
|------------|----------------|------------------------|
| 1 | N | 12 |
| | E | 0.9 |
| | SE | 8.5 |
| | S | 3.1 |
| | SW | 1.4 |
| | W | 6.8 |
| | NW | 26.2 |
| 2 | N | 2.3 |
| | E | 0.6 |
| | SE | 12.3 |
| | S | 0.3 |
| | SW | 0.9 |
| | W | 0.9 |
| | NW | 23.9 |

The temperature and relative humidity in each weather group is used to generate the values of fuel moisture content of 1, 10, and 100 h time-lag dead fuels classes, following the equations in [49]. The values of live herbaceous and live woody fuel moisture are directly imputed by the user. This information is stored in the fms file and later used in the fire behavior simulation.

The function *build_report* also creates a calibration report that briefly describes the study area, plotting its location and characterizing both the fire size distribution and inter-annual burned area variability (see Section S7.9 in Supplementary Materials). It also includes the weather classification, providing concise explanations of methods and figures (elbow method and Silhouette score) to assist the user in selecting both the clustering method and the final number of fire weather groups.

The function also exports a table with all the weather information. Hence, as an alternative to creating fire weather clusters and using them in the calibration process, the user can set specific weather scenarios (e.g., use extreme weather conditions or percentiles of temperature, wind speed, and relative humidity). For an alternative approach to weather analysis, the user must build a csv file input following the same data structure as the one created by the *build_report* function (see Table S1).

### 2.5. Defining the Number of Duration Parameters (Function buid_report)

The duration parameter (in minutes) sets the maximum simulation time for the fire spread calculations. Because MTT does not simulate fire suppression, fire spread stops when: (a) there are no burnable fuels (the fire spread encounters a barrier); or (b) when the simulation time reaches the maximum duration. Hence, the duration parameter plays a central role in the calibration process strongly affecting the fire size distribution. The fine-tuning of this parameter is required to replicate the historical fire size distribution, which can be the most challenging and time-consuming step of the calibration process. To calibrate the model, the user may need to define several classes of fire spread duration to replicate the historical fire size distribution (i.e., longer durations for larger wildfires; smaller durations for smaller wildfires), with different relative frequencies for classes of area. Therefore, not only it is challenging to define the duration of each class but also to define the number of duration classes to include in the simulations, as well as their relative weights.

MTTfireCAL can be used to define both the number of durations classes, as well as their specific values. First, the package identifies peaks in the historical fire size distribution using the function "peakdet" from the NADfinder package [50]. A duration class is set for each peak. Figure 3a shows an example for which peak identification recommended four duration classes for the model calibration, broadly corresponding to burned extents between 100 ha and 600 ha, from 600 ha to 1000 ha, from 1000 ha to 10,000 ha, and more than 10,000 ha. Alternatively, the user can manually set the duration classes. The output from the analysis of the duration parameter is also included in the calibration report. The definition of the duration classes will produce a table of relative frequencies (Table 3) that will be used later during the random generation of ignitions across the landscape (see Section 2.6: Generate Ignitions).

**Table 3.** Example of the relative frequencies (in percentage) of each duration class, as defined in Figure 3.

| Duration Class | Size Interval (ha) | Relative Frequency (%) |
|:---:|:---:|:---:|
| 1 | 100–600 | 70.7 |
| 2 | 600–1000 | 8.6 |
| 3 | 1000–10,000 | 17 |
| 4 | >10,000 | 3.7 |

Another key aspect of a calibrated fire spread model is its ability to reproduce the spatial distribution of wildfires. This feature is highly dependent on the ignition probability surface, on the fuel model map used, and on the duration values. The latter depends on the timeframe considered for the calibration, i.e., the years of the maps of fuel models. To assist the user in selecting which fuel model maps better represent the historical conditions associated with relevant wildfires, the *build_report* function exports the total burned area per year, which can be used to identify which was/were the most relevant year(s) for the overall burned area. The chosen year(s) is/are set to represent the historical fuel map(s) prior to the occurrence of a fire. Figure 3b shows that the years 2003 and 2018 represent ca. 80% of the total burned area between 2001 and 2022. Hence, when simulating the historical fire regime, at least two fuel maps representative of these years should be included to ensure that the spatial fire patterns are reproduced (see Section 2.8.2 and Section S7.3 in Supplementary Materials). If more than one fuel map is considered, weights must be

assigned to each based on their importance for the bulk of the burned area. Considering the former example, the fuel maps of 2003 and 2018 had weights of 0.6 and 0.4, respectively (Figure 3b).

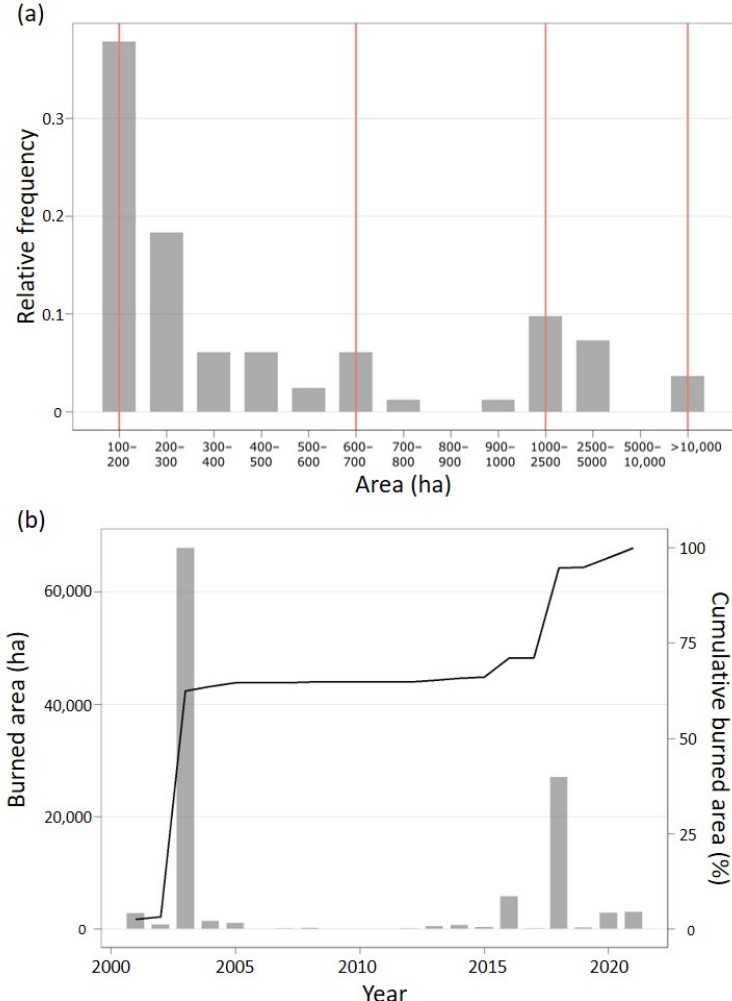

**Figure 3.** Example of (**a**) identification of peaks in the distribution of fire sizes (red lines); and (**b**) burned area distribution between 2001 and 2022. In panel a, the barplot represents the historical fire size distribution calculated from the fire perimeters given as input in the function *build_report* and each red line represents a different duration class.

*2.6. Generate Ignitions (Function gen_ign)*

The function *gen_ign* uses a surface probability grid to randomly sample ignition locations, i.e., areas with higher probability will have more ignitions. Random allocation of ignitions is also possible, but it is not recommended as it highly influences estimated fire size and burn probability [20].

In the function *gen_ign*, the user may specify the raster value for unburnable fuels (e.g., urban areas, water), ensuring that all ignitions are located in areas where a fire could potentially start. In some cases, the surface probability grid may show a zero probability of ignition in areas that have a burnable fuel type. To guarantee that an ignition may be placed in all burnable areas, the user may also set a new minimum ignition probability value. The function returns a point shapefile (optional) with all the ignitions generated and a text file with the ignition coordinates that will be used to run the FConstMTT. The total number of ignitions is defined by the user.

To reproduce the historical fire size distribution, one must ensure that the proportion of the different fire size classes is conserved. For instance, considering the fire size distribution in Figure 3a, the occurrence of a fire event that burns between 100 and 200 hectares is

ca. four times more likely than one that burns between 1000 and 2500 hectares. In other words, for one fire with an extent between 1000–2500 hectares, four other fires between 100–200 hectares occurred. The historical proportion between fire size classes is reproduced in the simulations by generating a different number of ignitions in each duration class. The same process is done for each fire weather cluster (if applicable), wind direction, and each fuel map considered (Equation (1)). Together, the combination of these factors forms a scenario with a given number of ignitions resulting from the product of the corresponding weights (the relative frequencies act as weights). The number of ignitions in each scenario is defined as follows:

$$Nign_j = R \text{ weather group} \cdot R \text{ WD}$$
$$W \text{ FM} \cdot R \text{ Dcl} \cdot \text{Total Nign} \tag{1}$$

where $Nign_j$ is the number of ignitions rounded to the units generated by the function *gen_ign* for the scenario *j*; *R* weather group is the relative frequency of the cluster or percentile considered; *R WD* is the relative frequency of the wind direction of each weather group; *W FM* is the weight of the fuel model map; *R Dcl* is the relative frequency of the duration class considered; and *Total Nign* is the total number of ignitions to be generated (sum of all scenarios).

For example, considering the relative frequencies shown in Table 2, the weather conditions associated with cluster 1 with the wind blowing from north (first row in Table 2), the fuel model map from 2003 (relative frequency of 0.6—Figure 3b), duration class 1 (representing fire sizes between 100 to 600 ha; Table 3) and a total number of ignitions equal to 5000, then:

$$Nign \text{ cluster } 1 \text{ N} = 0.120 \cdot 0.6 \cdot 0.707 \cdot 5000$$

$$Nign \text{ cluster } 1 \text{ N} = 254$$

A more comprehensive example is shown Table S2.

### 2.7. Running FConstMTT (Functions run_fconstmtt and run_fconstmtt_simple)

In the function *run_fconstmtt,* the user defines the range of duration values to be tested in a specific duration class, as well as the step used to set values. The range of duration values to be tested is subjective and depends on the user's intuition and experience. Using the example shown in Figure 3a, for each of the four duration classes, the user sets the "Minimum", "Maximum", and "Step" values. For instance, for the duration of class 1, we set the value of Minimum at 100 min, the value of Maximum at 200 min, and the Step value at 50 min. This results in three duration values to be simulated for this duration class (100 min, 150 min, and 200 min; Table 4). Following the same example, a total number of 336 combinations would be generated: 3 from duration class 1 × 7 from duration class 2 × 4 from duration class 3 × 4 from duration class 4.

**Table 4.** Example of a set of combinations given as input duration values to the *run_fconstmtt* function.

|  | Duration Class 1 | Duration Class 2 | Duration Class 3 | Duration Class 4 |
|---|---|---|---|---|
| *Minimum* | 100 | 250 | 600 | 1750 |
| *Maximum* | 200 | 400 | 900 | 2500 |
| *Step* | 50 | 25 | 100 | 250 |

The function *run_fconstmtt* creates all the input files and the batch file to run FConstMTT. One input file is created for each combination of one meteorological group, wind direction, fuel model, and fire spread duration (i.e., for each scenario). To run the simulations, FConstMTT executable must be available on the computer (can be downloaded at https://www.alturassolutions.com/FB/FB_API.htm) (accessed on 21 May 2023). The individual outputs are then stored with a unique name that allows us to trace them back to the scenarios they represent. Note that running the FConstMTT will most likely be the most

time-consuming step in the calibration process. Using a reasonable number of duration values to be tested helps speed up the process.

In cases where users have a priori knowledge of the environmental conditions during fire spread, or when a single set of weather conditions and duration is used (e.g., [51]), the function *run_fconstmtt_simple* can be used. However, this function is not detailed in this work.

### 2.8. Evaluating the Quality of the Calibration

2.8.1. Fire Size Distribution (Function *evaluate_fire_size*)

In the function *evaluate_fire_size*, the simulated fire size distribution is compared against the historical fire size distribution, and performance metrics are calculated. These include the linear Pearson correlation, the root mean square error (RMSE), the percentage of the normalized root mean square error (NRMSE) as implemented in R package forestmangr [52], the mean absolute error (MAE), the relative absolute error (RAE) as implemented in R package Metrics [53], and the Nash–Sutcliffe model efficiency (NSE) as implemented in R package ie2misc [54]. The equations used to calculate each metric can be found in Supplementary Materials Section S7.8.

In addition to these metrics, MTTfireCAL also creates a figure comparing the simulated fire size distribution using different combinations of duration values against the observed historical distribution (Figure 4 and Table 5). The creation of both quantitative statistics and figures allows us to thoroughly assess the quality of the calibration in reproducing the historical data [55]. By interpreting both outputs, it is possible to identify the combination of duration values that best replicates the historical fire size distribution. Nevertheless, the results should be used and interpreted with care. One should consider the quality and availability of the input data, and the level of accuracy which is required for the intended model application.

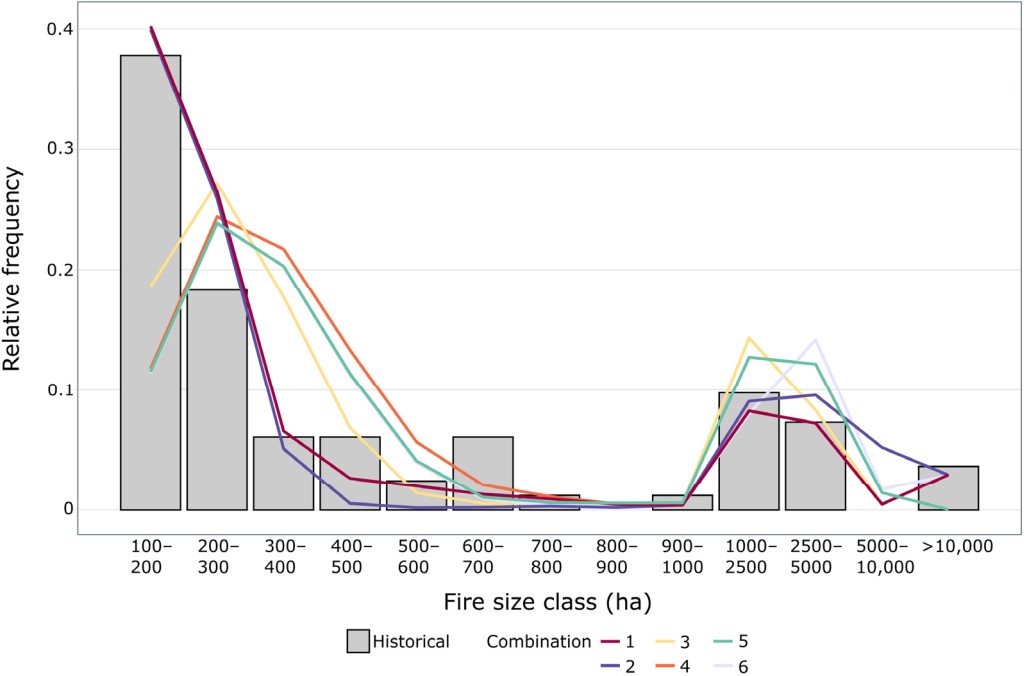

**Figure 4.** Example of fire size distribution generated by six combinations of duration values (lines) and their comparison to the historical fire size distribution (histogram). Similar figures are automatically exported after running the 'MTTfireCAL', namely, the function *evaluate_fire_size*.

**Table 5.** Performance metrics for each one of the combinations shown in Figure 4.

| Combination | NRMSE (%) | RMSE | Pearson Correlation | MAE | RAE | NSE |
|:---:|:---:|:---:|:---:|:---:|:---:|:---:|
| 1 | 38 | 0.029 | 0.974 | 0.019 | 0.281 | 0.91 |
| 2 | 47 | 0.036 | 0.955 | 0.027 | 0.409 | 0.87 |
| 3 | 91 | 0.070 | 0.722 | 0.043 | 0.650 | 0.5 |
| 4 | 116 | 0.089 | 0.510 | 0.051 | 0.775 | 0.19 |
| 5 | 116 | 0.089 | 0.519 | 0.056 | 0.849 | 0.2 |
| 6 | 116 | 0.089 | 0.508 | 0.055 | 0.829 | 0.19 |

In the example shown, combination 1 can be considered as the one that better reproduces the historical fire size distribution since it has the best associated performance, i.e., lowest RMSE, percentage NRMSE, MAE and RAE, and highest correlation and NSE (Table 5).

Note that the semi-automatic calibration using MTTfireCAL can also be an iterative process. It is possible that after running the FConstMTT for all the combinations generated, the calibration leads to unsatisfactory results. If this is the case, the user can repeat this process as many times as needed by readjusting the duration parameter(s) and/or changing the number of duration classes (Section 2.5). Nevertheless, one should bear in mind that increasing the number of combinations will lead to a larger amount of time spent running the fire spread simulations.

2.8.2. Burn Probability vs. Historical Fire Frequency (Function *evaluate_BP_nxburned*)

The burn probability is often compared against the historical fire frequency [8,22] to complement the capacity of the model to accurately reproduce the historical fire patterns. It is expected that a calibrated model shows a good correlation between the two variables, with areas that have higher fire frequency also having higher estimated burn probability (Figure 5).

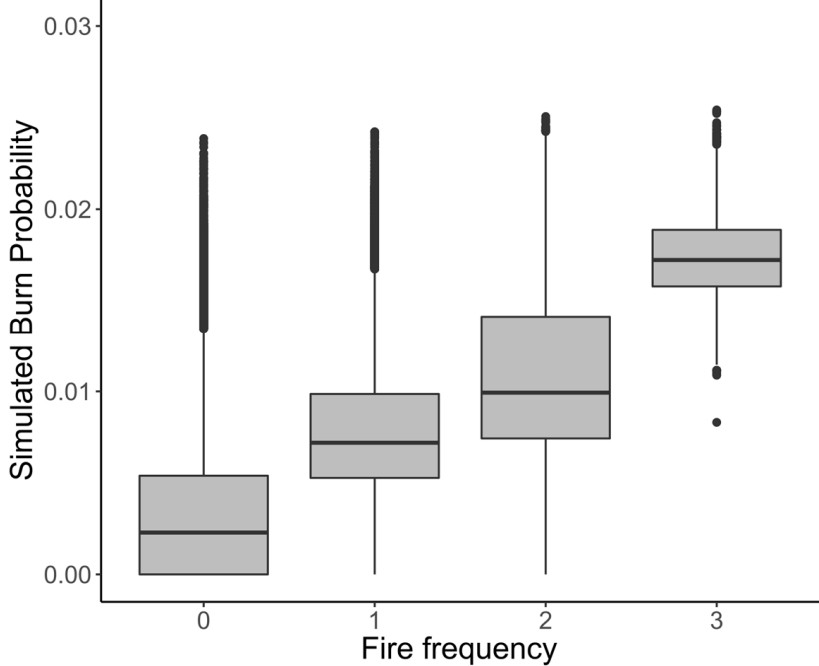

**Figure 5.** Example of boxplot showing the relationship between the historical fire frequency and the simulated burn probability. Similar figures are automatically exported after running the 'MTTfireCAL', namely, the function *evaluate_BP_nxburned*.

The burn probability is highly dependent on the ignition probability surface and the fuel model map (Figure S2). Hence, whenever the user obtains a weak correlation between the burn probability and the historical fire frequency, additional adjustments to the input data may be required to ensure a reasonable calibration. Changes to the ignition probability surface may imply including more years of data or filtering the data used (e.g., removing agricultural fires from the dataset and adding a threshold to the lowest burned area considered) or reprocessing the input data (e.g., breaking multi-day wildfires in single-day perimeters). Changes to the fuel map may imply including more fuel model maps that better represent the interval of years with the relevant burned area (see Figures S2 and S3). Other changes to both inputs might be necessary, depending on the study area and/or the work's objectives.

### 2.9. Minimum Number of Fire Runs Required for Calibration

To assess the minimum number of fire runs required for a trustworthy calibration, we used three Portuguese study areas with fire spread models previously calibrated, namely, the Barlavento Algarvio, the Médio Tejo, and the AM Porto (Figure 1).

Each one of the three landscapes was saturated with 200,000 ignitions. This corresponds to the baseline scenario. Then, from the pool of 200,000 simulated fire perimeters, we randomly sampled N fire runs with 10 replicates each, where N = 500; 1000; 5000; 10,000; 20,000; 30,000; 40,000; 50,000; 60,000; 70,000; 80,000; 90,000. The comparison between the fire size distribution resulting from a subset with N fires and the full simulated dataset was done by calculating the root mean square error (RMSE), percentage of the normalized root mean square error (NRMSE), MAE, RAE, and NSE, and by visually comparing the fire size distribution histograms.

We further anticipate that the minimum number of ignitions required for the calibration process will depend on the size of the landscape. To normalize the minimum number of fire runs required for calibration, we divided the total burnable area in each study area by the suggested total number of ignitions. The result is a ratio that is between the area (in hectares) per ignition. This ratio was then compared against the metrics listed above to create a rule of thumb that estimates the minimum number of ignitions required to calibrate the MTT.

## 3. Results

### 3.1. Minimum Number of Fire Runs Required for Calibration

Figures 6, S4 and S5 show the performance metrics when comparing the fire size distribution obtained from 200,000 fire runs and significantly fewer fire runs. All performance metrics showed a similar pattern across the study areas, despite the differences in the landscapes and historical fire regimes (Figures 6, S4 and S5). This pattern is characterized by an initial abrupt increase in the performance of the model, until reaching a plateau after ca. 20,000 fire runs (percentage NRMSE = $2.5 \pm 0.8$; RMSE = $0.0016 \pm 0.0005$; MAE = $0.0012 \pm 0.0003$; RAE = $0.0297 \pm 0.0158$; NSE = $1 \pm 0.002$).

Figures 7, S6, and S7 show the comparison of the histograms of fire size distribution obtained from 200,000 fire runs and significantly fewer fire runs. For all study areas, the variability around the median is relevant for many of the fire runs below 5000. This means that using little ignitions to calibrate the MTT can result in the misidentification of the optimal duration values. Altogether, the analysis of both the RMSE and histogram of fire size distribution point to a minimum required number of 5000 to 10,000 fire ignitions to reliably reproduce the historical fire size distributions. As expected, larger landscapes require (e.g., Médio Tejo) more fire runs.

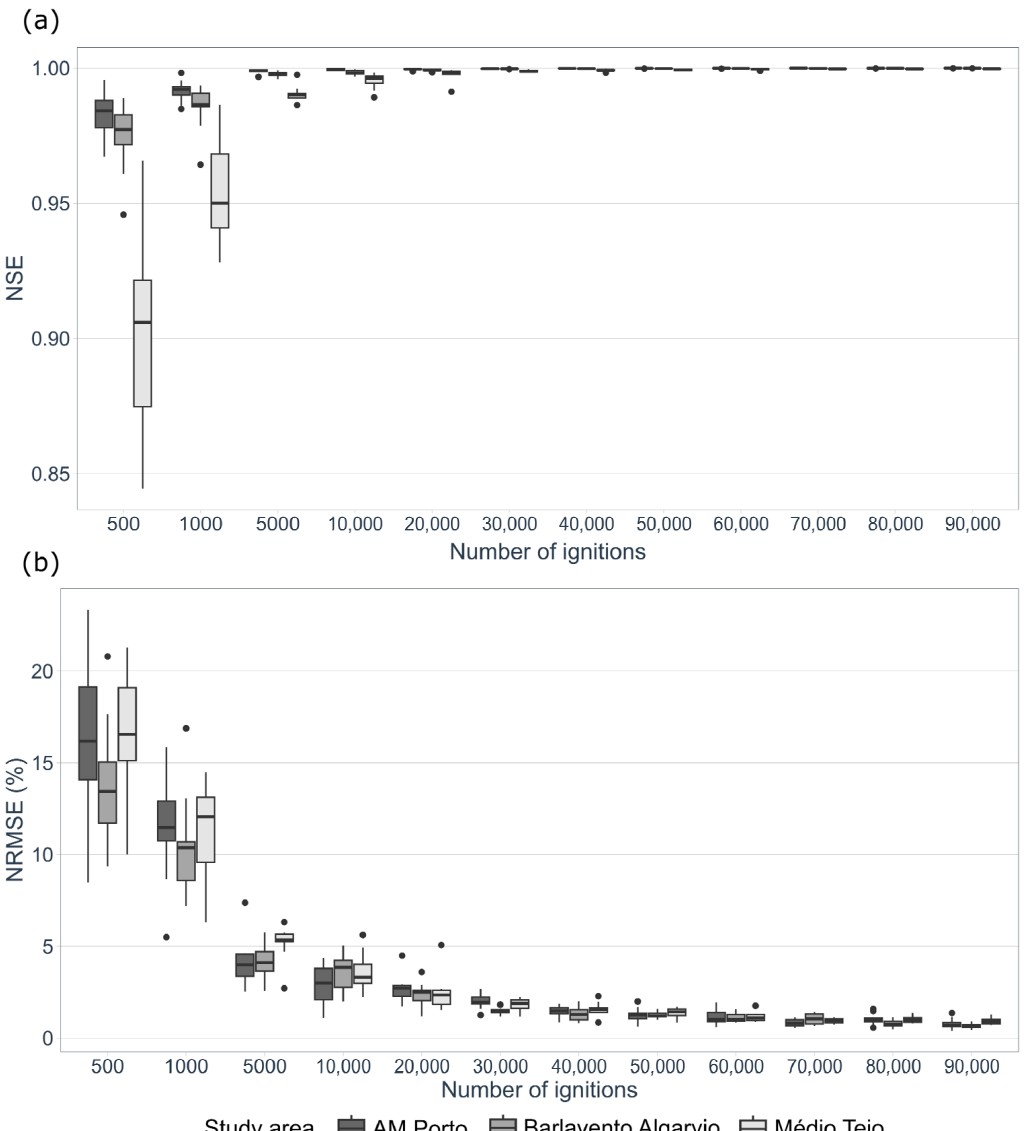

**Figure 6.** Nash–Sutcliffe efficiency (NSE); (**a**) percentage NRMSE (NRMSE); (**b**) calculated for the fire size distribution obtained from 200,000 fire runs and 500, 1000, 5000, 10,000, 20,000, 30,000, 40,000, 50,000, 60,000, 70,000, 80,000, 90,000 fire runs for AM Porto, Médio Tejo, and Barlavento Algarvio.

Figures 8, S8 and S9 show the analysis of the correlation between the burn probability estimated from the full simulation dataset and each subset of N fire runs. These figures show that using only 500 or 1000 fire runs leads to large variability in the estimated burn probability. This highlights the spatial dependency on the randomness of ignitions. When using 5000 fire runs, the pattern of estimated burn probability is similar to the one estimated for the full simulation dataset, with little variability between replicates. Similarly, to the fire size distribution, a minimum of 5000 to 10,000 fire runs are required to reliably reproduce the estimated burn probability.

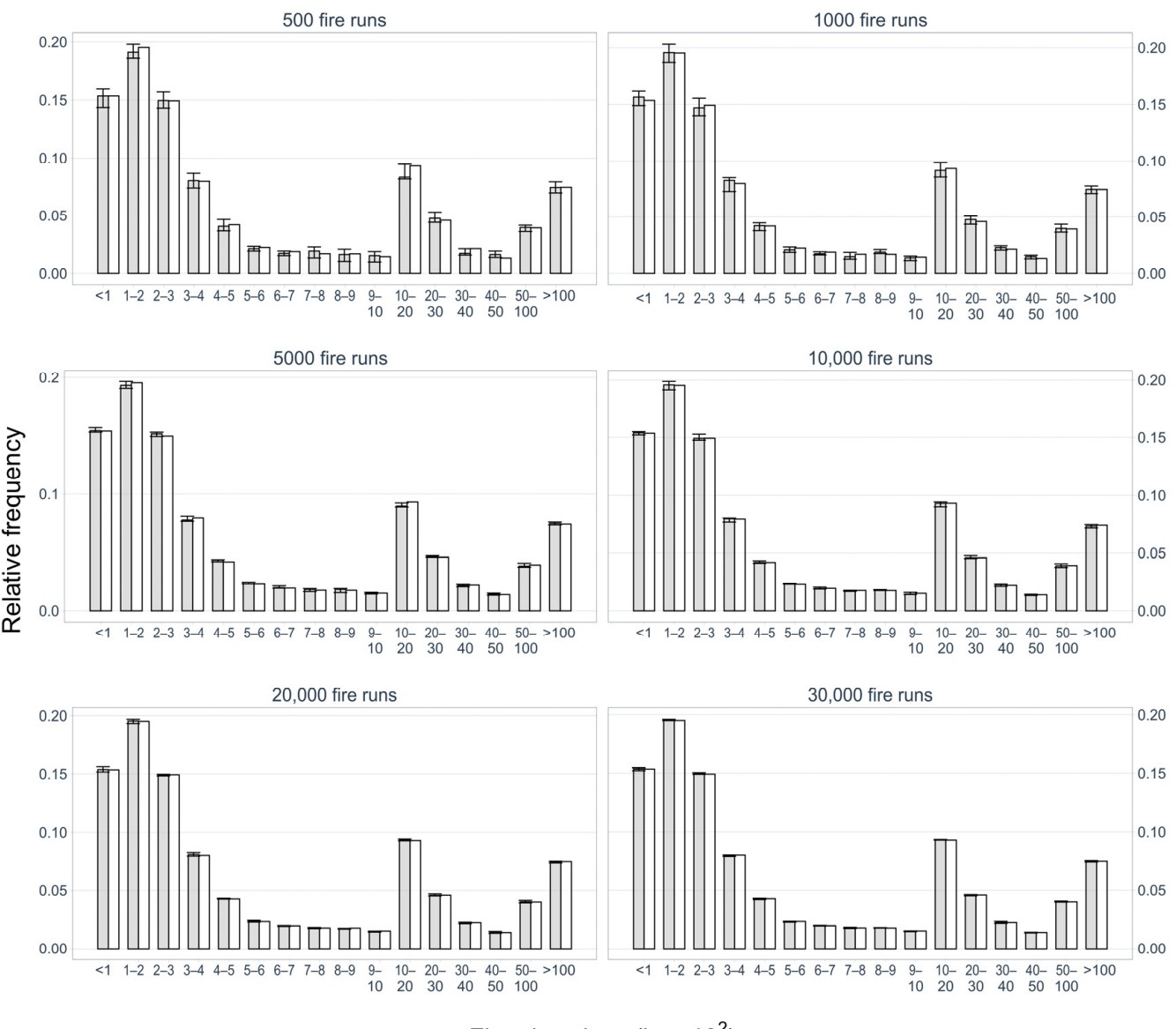

**Figure 7.** Fire size distribution for the baseline scenario calculated from 200,000 fire runs (white) and for the scenarios calculated from N fire runs (grey) for the Barlavento Algarvio study area. The error bar on top of the grey barplot represents the variability in the 10 replicates considered in each scenario. The grey bar represents the median value of the 10 replicates.

Figures S10 and S11 show the normalization of the number of ignitions with the burnable area in each landscape. Assuming a minimum number of ignitions required for calibration as N = 5000 ignitions for Barlavento Algarvio and AM Porto, and N = 10,000 for Médio Tejo, we divided the total burnable area in each study area by the corresponding number of ignitions. The results indicate that a ratio between 50 and 20 burnable hectares per ignition should be considered when setting the ignitions. Future applications can use this value as a benchmark. This ratio is applied to the study areas in Table S3.

Note that this analysis is valid only for calibration purposes. When using MTT fire spread model to produce fire behavior metrics and other relevant results (e.g., flame length, burn probability), the user must saturate the landscape with ignitions so that the total simulated burn area represents 10,000 fire seasons (e.g., [32]).

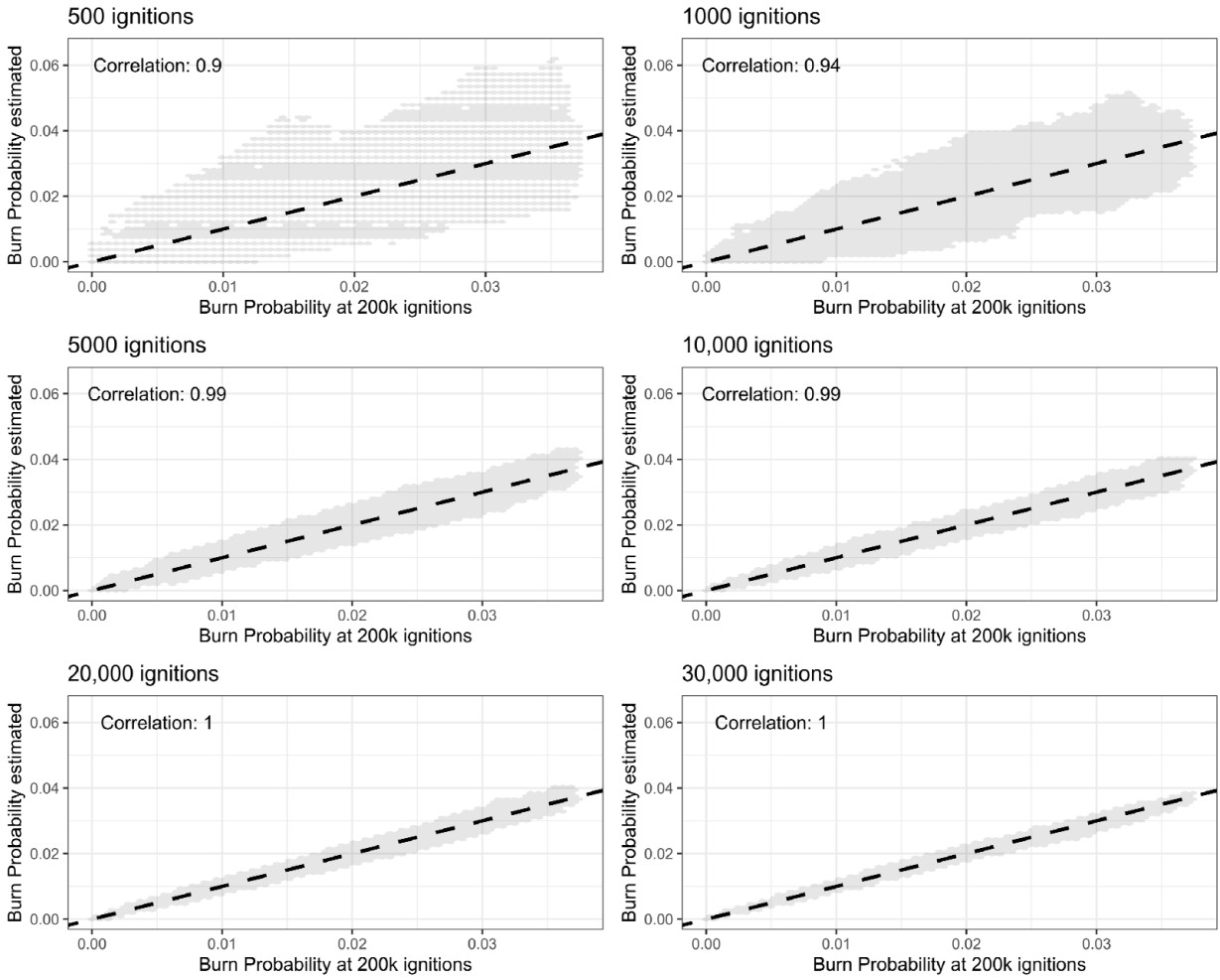

**Figure 8.** Correlation between the estimated burn probability calculated with 200,000 fire runs and the estimated burn probability calculated using 500, 1000, 5000, 10,000, 20,000, and 30,000 fire runs for the Barlavento Algarvio study area. Each scenario (except the 200,000 fire runs) has 10 replicates. The dashed line represents the 1:1 line. The top-left of each panel shows the Pearson correlation coefficient between the two variables.

*3.2. Validation*

To validate the use of MTTfireCAL and compare it against the typical manual calibration, we applied this procedure to two different study areas in Portugal: Médio Tejo and AM Porto (Figure 1). MTT fire spread model was previously calibrated for Médio Tejo [56] and AM Porto [57]. The calibration was carried out prior to the development of the MTTfireCAL R package and was conducted independently by two researchers.

We used MTTfireCAL to calibrate the MTT algorithm for the study areas of Médio Tejo and AM Porto, without any prior knowledge of the calibrated fire duration parameters. Overall, a total of ca. 37,000 combinations were evaluated for the study area of AM Porto and ca. 7500 for the Médio Tejo (Tables S4 and S5). The calibration process for AM Porto was done twice because the first set of combinations failed to satisfactorily reproduce the historical fire size distribution, resulting in a large number of combinations. The calibration process for each study area took approximately 3 h using an AMD Ryzen 9 3950X 16-Core Processor 3.49 GHz, with 32 GB of RAM. The entire calibration process in MTTfireCAL (from characterizing weather groups to evaluating the quality of the calibration) was completed in 3–4 days, which represents a decrease in one order of magnitude in the time dedicated to the calibration of the MTT when compared against the manual calibration that took more than 3 weeks (15 working days) in the study areas.

The two study areas showed a similar time required for the calibration, despite the number of combinations tested being five times larger for the AM Porto than for Médio Tejo. This was balanced by the fact that Médio Tejo is almost four times larger than AM Porto and the frequency of large fires (fire size > 1000 ha) is four times larger in the previous area. Hence, the overall computation time depends not only on the number of combinations tested but also on the size of the study area and the number of large fires (e.g., fire size > 1000 ha) simulated.

Figure 9 shows the comparison between the historical and simulated fire size distribution obtained with the manual calibration process and with the semi-automatic MTTfireCAL process. The fire size classes used in the comparison between historical and simulated distributions are the same as the ones used for manual calibration [56,57]. Overall, the simulated fire size distribution of both methods follows the historical distribution.

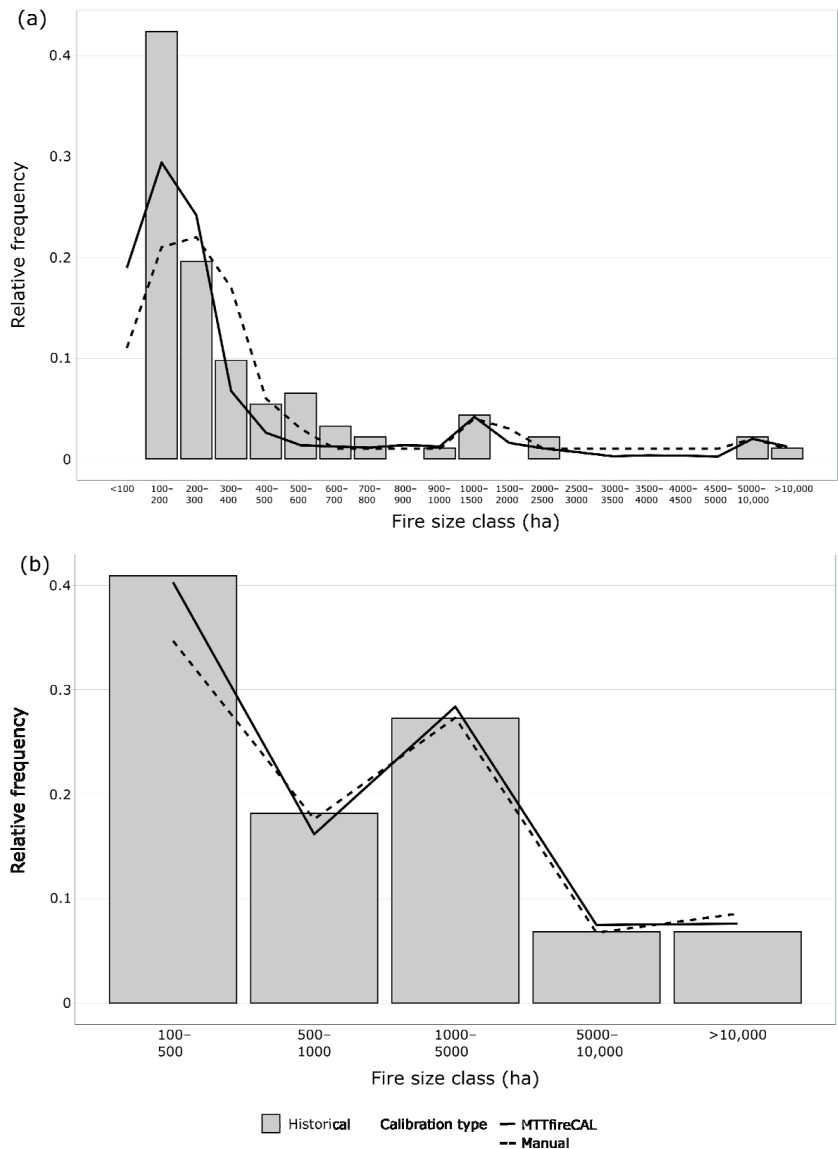

**Figure 9.** Comparison between the historical (barplots) and the simulated fire size distribution using the MTTfireCAL (solid line) and the manual calibration process (dashed line) for the study areas of AM Porto (**a**) and Médio Tejo (**b**). The classes of fire size distribution were kept the same as they were defined during the manual calibration process (prior to the development of MTTfireCAL) (see [56,57]).

Table 6 and S6 show the performance metrics of the calibrated MTT using MTTfireCAL and the manual trial-and-error calibration process for the three study areas. Both Médio Tejo and AM Porto show a decrease in the error (e.g., NRMSE decreased 34% for AM Porto and 9% for Médio Tejo) and an increase in model efficiency (e.g., NSE increased 0.17 for AM Porto and 0.04 for Médio Tejo). Hence, the calibration conducted using the MTTfireCAL showed better performance metrics than the manual calibration, suggesting that both faster and more accurate calibration can be obtained using the MTTfireCAL.

**Table 6.** Performance metrics for the calibrated MTT algorithm in the three study areas using MTTfireCAL (grey rows) and manual calibration (white rows).

| *Study Area* | NRMSE (%) | *Pearson Correlation* | *RAE* | *NSE* | *Spatial Correlation* |
|---|---|---|---|---|---|
| AM Porto | 68 | 0.95 | 0.33 | 0.87 | 0.4 |
| | 102 | 0.86 | 0.43 | 0.70 | 0.41 |
| Médio Tejo | 6 | 1 | 0.092 | 0.99 | 0.44 |
| | 15 | 0.99 | 0.15 | 0.95 | 0.38 |
| Barlavento Algarvio | 38 | 0.97 | 0.281 | 0.91 | 0.59 |

The performance metrics for the Barlavento Algarvio study area correspond to combination 1 shown in Figure 4. The performance metrics were calculated using 5000 fire runs for the MTTfireCAL and 200,000 fire runs for manual calibration. NRMSE (%), Pearson correlation, RAE, and NSE correspond to the comparison between simulated and historical fire size distribution, while the spatial correlation corresponds to the comparison between historical fire frequency and estimated burn probability. The full performance metrics are shown in Table S6.

The performance metrics were found to be highly dependent on the number of fire size classes used for the calibration (i.e., the level of detail). For instance, Médio Tejo was calibrated considering only five fire size classes and shows the best values for the performance metrics (i.e., lowest percentage NRMSE, RMSE, and RAE, and highest correlation). On the other hand, AM Porto was calibrated considering a total of 20 fire size classes and shows the poorest performance (i.e., highest percentage NRMSE, RMSE, and RAE, and lowest correlation). However, when applying the five fire size classes of Médio Tejo to AM Porto, the performance metrics revealed an increase in the quality of both the manual and MTTfireCAL calibrations to values similar to Médio Tejo (see Figure S12 and Table S7). In future applications, users may rely on these performance values to benchmark their calibration procedures but should acknowledge the influence that the level of detail (i.e., number of fire size classes) has in the metrics.

## 4. Discussion and Conclusions

In this work, we present the new R package "MTTfireCAL", an innovative tool to assist in the calibration of the MTT fire spread model, from the characterization of the study area and fire weather conditions to the evaluation of model performance using different parameters. To demonstrate the usefulness of MTTfireCAL, we applied it to one study area in Portugal and validated the quality of the semi-automatic calibration using two other study areas with different fire regimes, also in Portugal. Overall, the use of the MTTfireCAL R package allowed for a faster and better calibration of the MTT fire spread model when compared with the typical trial-and-error calibration. Furthermore, we provided the performance values of each of the calibrated MTT models, which can be used to benchmark future calibration procedures.

With this study, we show that:

- The minimum number of fire runs (or ignitions) required to reproduce the historical fire patterns during the calibration is dependent on the size of the landscape;
- We suggest a value between 50 and 20 for the ratio between the burnable area in the landscape (in hectares) and the number of ignitions used in the calibration can be used as a rule of thumb to assess the minimum number of ignitions required for calibration;
- The combination of both the MTTfireCAL tool and a low number of ignitions used resulted in a faster and better calibration than the manual trial-and-error process, reducing the amount of time required to calibrate the MTT in one order of magnitude;
- Because MTTfireCAL runs multiple combinations automatically, it releases the user to complete other tasks while calibrating the MTT.

We are confident that this tool will be of great interest to the academic and operational community working with MTT fire spread simulations. MTTfireCAL has great potential to support better fire management and research, particularly in the areas of hazard and risk reduction, and hence, better support the design of fuel reduction strategies. MTTfireCAL can assist and guide new users into a fast and high-quality calibration. Notwithstanding, one should consider that "insight, intuition and sound judgement play an important role" in the modeling process [55], particularly when assessing the quality of the model.

*Future Work*

The MTTfireCAL R package will be continuously updated following methodological advances in fire spread modeling and will evolve in response to the needs of a growing global community of users. The github of MTTfireCAL (https://github.com/bmaparicio/MTTfireCAL, (accessed on 21 May 2023)) will feature regular updates in both the functions and documentation (including tutorials).

Future improvements will include (i) the implementation of parallel processing in all the functions, and the addition of new fire weather clustering methods (e.g., density-based clustering); (ii) the possibility of downloading and using other meteorological data sources to characterize fire spread besides ERA5-Land; and (iii) new methods to calculate dead and live fuel moisture content. New MTT calibration methods may be added as new data becomes available. One key feature in calibrating and validating fire behavior model outputs is its comparison against observed fire metrics, such as rate of spread and fireline intensity [58], as the current calibration process solely focuses on reproducing historical fire size and frequency. Although comprehensive open-access fire behavior data is difficult to obtain, new fire behavior datasets are being published [59,60], which can foster the use of fire behavior metrics in the calibration of MTT models.

Finally, we plan to include new functions to generate and manipulate landscape files within the R package, so that MTTfireCAL becomes completely independent from FlamMap. Regarding the outputs, a future R package will be developed to build important metrics of fire behavior such as conditional flame length, annual burn probability, fire potential index [21], or the high-intensity burn probability and high flame length probability [61]. Altogether, the planned new functions will allow us to further expand the utility of MTTfireCAL, as the user will be able to calibrate and both rapidly assemble and analyze multi-scenario fire behavior outputs.

MTTfireCAL and all the documentation, manuals, and tutorials are freely available at: https://github.com/bmaparicio/MTTfireCAL (accessed on 21 May 2023).

**Supplementary Materials:** The following supporting information can be downloaded at: https://www.mdpi.com/article/10.3390/fire6060219/s1.

**Author Contributions:** B.A.A.: conceptualization, methodology, software, validation, formal analysis, investigation, data curation, writing—original draft, and visualization. A.B.: conceptualization, validation, writing—review and editing, and supervision. J.M.C.P.: writing—review and editing and supervision. A.C.L.S.: conceptualization, validation, writing—review and editing, and supervision. All authors have read and agreed to the published version of the manuscript.

**Funding:** This research was supported by the Forest Research Centre research unit, funded by Fundação para a Ciência e a Tecnologia I.P. (FCT), Portugal (UIDB/00239/2020), and by the project FRISCO: Managing Fire-induced Risks of Water Quality Contamination (Ref. a PCIF/MPG/0044/2018), and FIRE-MODSAT II (no. PTDC/ASP-SIL/28771/2017), also funded by FCT. Bruno Aparício was supported by the Ph.D. fellowship funded by FCT (UI/BD/150755/2020). Ana Sá was supported under the framework of contract program no. 1382 (DL 57/2016/CP1382/CT0003). Akli Benali was funded by FCT through a CEEC contract (CEECIND/03799/2018/CP1563/CT0003).

**Institutional Review Board Statement:** Not applicable.

**Informed Consent Statement:** Not applicable.

**Data Availability Statement:** Data is available at https://github.com/bmaparicio/MTTfireCAL (accessed on 21 May 2023).

**Acknowledgments:** The authors thank Luiz Felipe Galizia for his valuable comments and feedback on an earlier version of the MTTfireCAL. We are also grateful to Chiara Bruni and Beatriz Lourenço for providing the manually calibrated MTT models for the study areas.

**Conflicts of Interest:** The authors declare no conflict of interest.

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
