# Peer review of "MTTfireCAL Package for R—An Innovative, Comprehensive, and Fast Procedure to Calibrate the MTT Fire Spread Modelling System"

_fire, doi:10.3390/fire6060219_

Round 1

Reviewer 1 Report

Dear Authors,

This manuscript presented the “MTTfireCAL” package for R, an instrument enabling fast calibration of the MTT fire spread models. Also, it expressed the main methodological stages. Additionally, it predicted the minimum number of fire runs needed to ensure a reliable calibration. Generally, although the authors have done their best to present a novel matter, the manuscript suffers from some structural errors. In terms of conception, they managed to present a novel research in the field. Before publication, following comments should be addressed accurately.

-Abstract is so vague. Passive sentences should be used. It should be comprehensive and brief.

-Old references in introduction should be updated. The authors should use further updated references particularly from the journal. Recent works should be added.

-The manuscript should be edited grammatically. Also, the similarity index of the manuscript should be lower than 20%.

-The range of the selected parameters should be justified. Why such ranges did you consider? Also, the quality of the figures is low.

-Explain more about the novelty of the manuscript particularly in the conclusion using bullet points.

Regards

There are many English language errors that should be addressed.

Author Response

Reviewer 1

This manuscript presented the “MTTfireCAL” package for R, an instrument enabling fast calibration of the MTT fire spread models. Also, it expressed the main methodological stages. Additionally, it predicted the minimum number of fire runs needed to ensure a reliable calibration. Generally, although the authors have done their best to present a novel matter, the manuscript suffers from some structural errors. In terms of conception, they managed to present a novel research in the field. Before publication, following comments should be addressed accurately.

-Abstract is so vague. Passive sentences should be used. It should be comprehensive and brief.

A: Thank you for this comment. We agree and we've changed the abstract. It is now more comprehensive.

-Old references in introduction should be updated. The authors should use further updated references particularly from the journal. Recent works should be added.

A: Thank you for this comment. Old references refer to the fundamentals of the calculation of fire behaviour and Minimum Travel Time. We do not believe newer ones should replace these references. Also, references to recent works are already present in the introduction (e.g. Ager et al., 2017; Sá et al., 2022; Alcasena et al., 2019; Salis et al., 2021; Palaiologou et al., 2020; Oliveira et al., 2016; Alcasena et al., 2022; Galizia et al., 2021; Ager et al., 2021; Palaiologou et al., 2021).

-The manuscript should be edited grammatically. Also, the similarity index of the manuscript should be lower than 20%.

A: We detected and corrected some typos in the text. We also changed many sentences for clarity.

-The range of the selected parameters should be justified. Why such ranges did you consider? Also, the quality of the figures is low.

A: We’ve added the following explanation about the choice of the ranges to test: The range of duration values to be tested is subjective and depends on the user’s intuition and experience.

Regarding the quality of the figures, high-quality figures will be updated after acceptance.

-Explain more about the novelty of the manuscript particularly in the conclusion using bullet points.

A: Thank you for this comment. We added bullet points to the conclusion.

Reviewer 2 Report

The paper presents an R package for the calibration of the MTT fire spread model. It is clearly written an explains the usage of each function concisely. The example calibrations and validation lend support to the model and provide a prospective user with adequate information to apply the package to their own calibration needs and clearly shows the benefits of this approach over a manual one. This package obviously has strong utility for MTT calibration, but some of the functions (get_fire_weather, for example) could also be used in other fire modeling work and therefor the paper has broader relevance to the fire science community. Overall, I think this is a valuable contribution and I see no issues with publishing the manuscript in its current form.

Author Response

Thank you for your comments

Reviewer 3 Report

This manuscript presents the MTTfireCAL package for R, a tool to calibrate fire duration when running MTT. I think the R package is a fantastic idea with multiple possibilities. The manuscript is correctly written and provides a clear description on how the user should proceed. My only concern is that authors sometimes mention that this calibration process allow to reproduce fire behaviour (from line nº 54 to 60) BUT, since what it is adjusted is the fire duration to recreate historical fire size, fire behaviour characteristic such as rate of spread could be (and most likely) totally altered and not reliable. I just only suggest that the author should underline this in the abstract, introduction, discussion and conclusions to avoid misinterpretations from non-experts.

Some minor suggestions:

L16: add      … “to reproduce observed fire regimens”

L217: It could be very useful for the user if you add a very brief description or references on how to proceed to create the ignition density layer

L225: Here you can also add some available datasets, for instance,  EFFIS fuel map or the new one released by Aragoneses et al 2023.

L436: I think there is a problem with an undesired table

L556: days and weeks do not correspond

L607: the link does not work

Author Response

This manuscript presents the MTTfireCAL package for R, a tool to calibrate fire duration when running MTT. I think the R package is a fantastic idea with multiple possibilities. The manuscript is correctly written and provides a clear description on how the user should proceed. My only concern is that authors sometimes mention that this calibration process allow to reproduce fire behaviour (from line nº 54 to 60) BUT, since what it is adjusted is the fire duration to recreate historical fire size, fire behaviour characteristic such as rate of spread could be (and most likely) totally altered and not reliable. I just only suggest that the author should underline this in the abstract, introduction, discussion and conclusions to avoid misinterpretations from non-experts.

A: Thank you for this comment. We agree that this must be clear. We clarified the abstract, introduction and discussion and conclusion. The sentence highlighted by the reviewer now reads: Afterwards, the MTT algorithm needs to be calibrated to ensure that the estimated fire patterns are reliable (Cardil et al., 2021). Failing in doing so may lead to errors in reproducing key fire descriptors, such as burn probability (Massada et al., 2011), ultimately undermining the use of fire simulation for research and management purposes.

In the discussion and conclusion we also added: One key-feature in calibrating and validating fire bahavior model outputs is its comparison against observed fire metrics, such as rate of spread and fireline intensity (Cardil et al., 2019), as the current calibration process solely focuses on reproducing historical fire size and frequency.

Some minor suggestions:

L16: add      … “to reproduce observed fire regimens”

A: We added it accordingly.

L217: It could be very useful for the user if you add a very brief description or references on how to proceed to create the ignition density layer

A: Thank you for this comment. We changed the paragraph and add this information. The paragraph now reads:

The ignition location is an essential input to estimate fire spread and behaviour descriptors, particularly burn probability (Massada et al., 2011), as it sets the starting point of the fire spread. The location of ignitions used to simulate fire spread is derived from an ignition probability surface that reproduces the broad historical spatial ignition patterns in the study area. Usually, the ignition probability surface is created from the historical ignition points by creating a smooth grid using a fixed search distance (e.g. kernel density; Benali et al., 2021).

L225: Here you can also add some available datasets, for instance,  EFFIS fuel map or the new one released by Aragoneses et al 2023.

A: Thank you for this comment. We added the two references when discussing the custom fuel maps. It reads: If custom fuels are used (e.g. Aragoneses et al., 2023; EFFIS 2017; Fernandes et al., 2009), a fuel model file containing their parameterization is required (.fmd file).

L436: I think there is a problem with an undesired table

A: There seems to be a problem with the formatting of Fire. The floating row is part of table 6 in the word document. We inserted the Table again and hopefully the problem is solved.

L556: days and weeks do not correspond

A: We meant working days. Thank you for noticing this. We changed accordingly.

L607: the link does not work

A: The problem is with the formatting of Fire. The link is https://github.com/bmaparicio/MTTfireCAL but with the formatting it becomes https://github.com/bmapari-, which is not correct.